# The Four Ws of the Fourth Dose COVID-19 Vaccines: Why, Who, When and What

**DOI:** 10.3390/vaccines10111924

**Published:** 2022-11-14

**Authors:** Ka-Wa Khong, Ruiqi Zhang, Ivan Fan-Ngai Hung

**Affiliations:** 1Department of Medicine, Li Ka Shing Faculty of Medicine, University of Hong Kong, Hong Kong, China; 2State Key Laboratory for Emerging Infectious Disease, Li Ka Shing Faculty of Medicine, University of Hong Kong, Hong Kong, China; 3Carol Yu Centre for Infection, Li Ka Shing Faculty of Medicine, University of Hong Kong, Hong Kong, China

**Keywords:** COVID-19 vaccine, booster dose, Omicron variant

## Abstract

With the emergence of SARS-CoV-2 variants, vaccine breakthrough is a major public health concern. With evidence of reduced neutralizing antibody activity against Omicron variants and fading antibody level after the third-dose booster vaccine, there are suggestions of a fourth-dose booster vaccine. In this review, the benefits of a fourth-dose booster is evaluated from four perspectives, including the effectiveness of the booster dose against virus variants (Why), susceptible groups of individuals who may benefit from additional booster dose (Who), selection of vaccine platforms to better enhance immunity (What) and appropriate intervals between the third and fourth booster dose (When). In summary, a fourth dose can temporarily boost the immune response against SARS-CoV-2 variants and can be considered for specific groups of individuals. A heterologous vaccine strategy using mRNA vaccine in individuals primed with inactivated vaccine may boost immunity against variants. The timing of the fourth dose should be individualized but an interval of 4 months after the third-dose booster is appropriate. A universal fourth booster dose is not necessary.

## 1. Introduction

SARS-CoV-2 is the coronavirus causing the COVID-19 pandemic. Since the introduction of the COVID-19 vaccine, billions of doses of vaccine over different platforms have been administrated. Serum neutralizing antibody level is used to assess vaccine immunogenicity and efficacy; hence, with evidence of a declining antibody level a few months after vaccination or infection [1], a third booster dose was proposed and adopted in many countries. Furthermore, the emergence of the Omicron variant from the end of 2021 raises great concerns about vaccine breakthrough [2], as multiple studies have demonstrated that this new variant of concern (VOC) can escape antibody neutralization from the serum of both previously infected patients and vaccinated individuals [3,4,5]. In view of the substantial functional and structural differences between the Omicron variant and other previous variants leading to ineffective neutralizing antibodies, it was suggested that Omicron could be considered as a distinct SARS-CoV-2 serotype [6]. Interestingly, serum from recipients of a third booster dose, when compared to individuals who did not receive the third dose, showed better neutralization activity against the Omicron variant [7,8], supporting the administration of a third dose of vaccine. Now, the debate continues as to whether a fourth booster dose should be given. Some believe that a fourth-dose booster can help maintain a high serum neutralizing antibody level which protects against infection, while some worry about vaccine equity, arguing that the repeated booster dose strategy of high-income countries prevents the low- and middle-income countries from vaccinating their most vulnerable populations [9,10]. In this review, we aim to assess the current evidence of booster doses against variants and discuss the important considerations we should think about: Why, Who, When and What.

## 2. Why Is the Fourth-Dose Booster Necessary

Currently, a third-dose booster is universal in some parts of the world but not others. Up to 31 October 2022, the number of doses of COVID-19 vaccine boosters administered is highest in Chile, with 139.3 doses of booster dose given per 100 individuals. The number is lower in India and Russia, with 15.5 doses and 13.2 doses of booster dose given per 100 individuals. Overall, the world average dose of booster per 100 people is 32.5 [11]. As mentioned in the previous section, one major concern is whether the currently used vaccine, developed with the Wild-Type strain, is effective against variants such as Omicron, as there were reports of breakthrough infection with Omicron despite an mRNA vaccine booster dose [12]. Moreover, it was also reported that vaccine efficacy of the third dose started to decline from two to four months after vaccination [13]. The vaccine effectiveness was estimated to decline from around 70% a week after the booster dose to around 40% at 15 weeks or more after the booster dose against both BA.1 and BA.2 sub-lineages [14]. In fact, even though the third booster dose can induce neutralizing antibodies against the Omicron variant, the antibody titre is non-comparable to other virus strains, including the Wild-Type, Beta and Delta variants [8,15,16]. Unlike influenza, against which a hemagglutination inhibition (HI) assay titre of 40 is defined as seroprotective [17], currently there is no definition of seroprotective antibody titre against COVID-19; hence, it is difficult to conclude whether the neutralizing antibodies induced by the third dose are adequate. Moreover, the question of how long the neutralization antibody response against the Omicron variant can persist after the third dose is still under investigation. In conclusion, it is challenging to determine whether a third booster dose is protective solely based on the presence of neutralizing antibodies. Beside humoral immunity, cellular immunity is also vital in viral infections. T cell response was suggested to be important for protection against severe COVID-19 infection [18]. In contrast to serum-neutralizing antibodies, a longitudinal study found that T cell responses could still be detected in convalescent individuals 12 months after initial infection [19]. Moreover, studies have demonstrated that T cell responses to Omicron are preserved in most infected or vaccinated individuals, and an additional booster vaccine can further enhance effector T cell responses to the Omicron variant [20,21]. Recently, in vitro studies demonstrated that a fourth dose of COVID-19 vaccine can boost cellular and humoral immunity, and the peak responses were similar to the peak responses after the third dose, unlike the superiority of the third dose to the second dose [22,23] (Figure 1).

The above-mentioned results are based on laboratory experiments. To determine the effectiveness of booster-dose vaccination it is also important to consider real life numbers. It was reported that fully vaccinated and boosted (three doses) individuals, when compared to those who are fully vaccinated (two doses) and unvaccinated, had a better protection against COVID-19 infection by the Omicron and Delta variants based on test-negative case–control analysis and lower in-hospital mortality rates [24,25]. In addition, according to the Morbidity and Mortality Weekly Report (MMWR) from the US Department of Health and Human Services, during the period of Delta and Omicron variant emergence, individuals who received the third dose had a better protection against hospitalization and death. A similar result was also observed in Hong Kong; individuals who received a third booster dose were protected against severe disease and death within 28 days of a positive test [26]. One might argue that the fully vaccinated individuals may have received the vaccines for more than 6 months and the declining antibody level lead to the difference; nevertheless, among individuals who visited the emergency department because of COVID-like illness, there were also additional benefits of a third booster dose compared to individuals who had just completed their first two doses within 180 days, indicating that a third booster dose does more than just boosting the antibody level [27,28]. Despite the enhanced protection, the effect of the third dose vaccine cannot last for a long time against Omicron. Andrews at al. found that the effectiveness of three doses of the BNT162b2 vaccine against Omicron declined to 45.7% at ≥10 weeks from 67.2% at 2–4 weeks after the third dose [29]. One clinical study reports similar results that the third dose of mRNA-1273 showed 71.6% effectiveness against Omicron at 14–60 days and then declined to 47.4% at >60 days [30]. In addition, the administration of a fourth dose of COVID-19 vaccine, compared to individuals who only received the third-dose booster, was demonstrated to prevent severe outcomes and reduced risk of death among residents in long-term care facilities [31,32]. Thus, the fourth dose of COVID-19 vaccine did induce additional protections.

## 3. Who Should Take the Fourth Dose Booster

After exploring the efficacy of booster doses against the Omicron variant, the next question to ask is which group of individuals can benefit from the extra shots. Three unique groups of people are of interest, including elderlies, immunocompromised patients and individuals who were previously infected with SARS-CoV-2 (Figure 2).

As people age, there is a progressive deterioration of the immune system response, known as immunosenescence, and consequently there may be poorer results of vaccinations [33]. The booster-dose vaccine was demonstrated to induce robust cellular and humoral immunity against SARS-CoV-2 in older adults aged more than 80 years old and previously unresponsive elderlies [34], and the protection is supported by the lower risk of infection, hospitalization, ICU care and death [35,36]. Additionally, studies from Japan using mRNA vaccines demonstrated that SARS-CoV-2 antibody titer may decline faster in older individuals, evidenced by lower titer at 6 months after vaccination [37,38]. Therefore, an additional booster dose may be needed to mount a protective immune response in elderlies.

Patients can be immunocompromised due to different etiologies, ranging from cancer to recipients of solid organ transplant. A literature review suggested that immunocompromised hosts may be at increased risk of severe COVID-19 disease and death [39], and the vaccine response in these individuals can be poor [40,41]. A booster dose was demonstrated to be effective against Omicron in cancer patients; patients with autoimmune diseases including rheumatic diseases, inflammatory bowel diseases, and vasculitis; and patients taking biologics, such as Rituximab and Ocrelizumab [42,43,44,45,46,47,48]. Robust COVID-19 vaccination responses were also observed in allogenic stem cell recipients [49,50]; nevertheless, immunogenicity of the vaccine among solid organ transplant recipients is controversial. For these patients, the fourth booster dose of vaccine induced some neutralizing antibody response against the Omicron variant, but the level of protection was questioned, as Thomson et al. reported that a significant proportion of solid organ transplant recipients remained seronegative and had poor T-cell responses after the third and fourth doses of SARS-CoV-2 vaccines [51,52,53]. Individuals with impaired B cell and T cell function may avoid development of severe COVID-19 symptoms, but at the same time fail to clear the virus, leading to a chronic SARS-CoV-2 carrier state [54]. In immunocompromised individuals who demonstrated suboptimal vaccine response, instead of repeated vaccination, an alternative approach could be the injection of monoclonal antibody cocktails, such as Evusheld, which demonstrated neutralizing activity against BA.1 and BA.2 sub-lineages [55].

Whether the vaccine recipient is previously infected with SARS-CoV-2 or naive to the virus is important, as a study had revealed that vaccination can induce a distinct immune response in previously infected individuals when compared to naïve individuals. There are more memory B cells, neutralizing antibodies and a distinct population of CD4 T cells that express interferon (IFN)-gamma and interleukin (IL)-10 [56]. On the other hand, since the emergence of the Delta and Omicron variants, there are also a group of individuals who were infected after vaccination, and it was demonstrated that neutralizing antibody titers after Omicron BA.1 breakthrough infection is high [57]. Interestingly, vaccination may also benefit patients with long COVID symptoms, as systemic reviews have suggested that vaccination before SARS-CoV-2 infection can reduce incidence of long COVID symptoms, while vaccination after an acute infection may also alleviate the long COVID symptoms [58,59]; however, these data were based on individuals who received two doses of vaccine. More data are needed to determine whether individuals who were infected may benefit from a fourth-dose booster in terms of protection and long COVID symptoms.

**Figure 2 vaccines-10-01924-f002:**
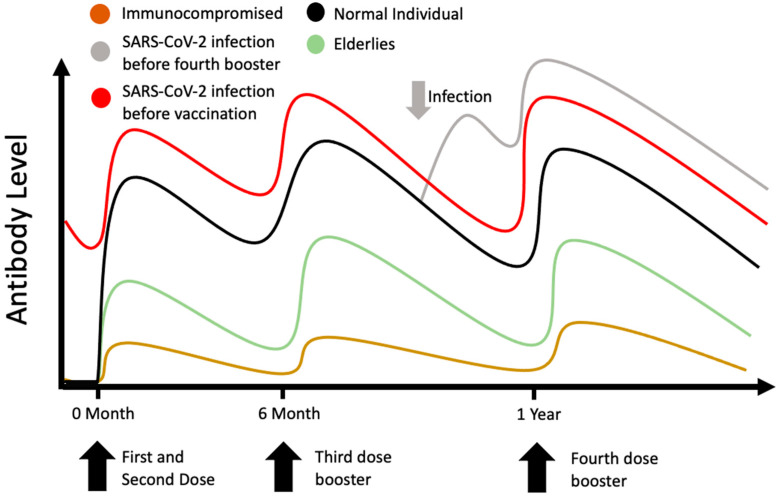
Changes in serum antibodies after vaccination or infection in different patient groups. Elderly patients may have a poorer response to vaccination and lower titre of serum-neutralizing antibodies after vaccination [33,37,38]. Immunocompromised patients, especially patients with solid organ transplants, may have suboptimal response to vaccination with a low level of neutralizing antibodies or being seronegative after the third and fourth booster doses [40,41,51,52,53]. Individuals who were infected before vaccination or have breakthrough infection after vaccination were demonstrated to have a higher level of neutralizing antibodies [56,57].

## 4. What Vaccine Should Be Used as the Fourth-Dose Booster

The vaccine platform of the fourth dose may also determine the immune response against virus variants. Some of the currently approved vaccines platforms include the inactivated virus vaccine (CoronaVac), mRNA vaccine (BNT162b2, mRNA1273), viral vector vaccine (ChAdOx1 nCoV-19, AD5-nCoV) and subunit vaccine [60,61,62,63,64,65]. Previously, a heterologous vaccine strategy was proposed for the third-dose booster and was shown that by giving vaccine of different platforms, immunogenicity against SARS-CoV-2 variants was enhanced (Table 1). The benefits of using a heterologous vaccine strategy are most prominent in individuals who received two doses of inactivated vaccine as the prime dose, as evidenced by a higher tier of antibody against virus variants [8,66,67,68,69]. In addition, a heterologous vaccine strategy is associated with higher binding affinity and increased breadth of reactivity against SARS-CoV-2 variants [70]. As a result of quantitatively and qualitatively enhanced antibody response, a heterologous boosting was demonstrated to be associated with lower infection rate [71]. Furthermore, dosage of the booster dose may also affect the vaccine response; studies suggested that the mRNA-1273 vaccine has a higher immunogenicity and effectiveness as compared to the BNT162b2 vaccine [72,73]. Using mRNA-1273 as the third-dose booster was also demonstrated to be superior than BNT162b2 in individuals primed with the BNT162b2 vaccine [67]. One possible explanation could be a higher vaccine dose, as one dose of mRNA-1273 contains 100 μg of active substance while one dose of the BNT162b2 vaccine contains 30 μg of active ingredient [61,62]. The dosage of the vaccine may be another consideration when considering what vaccine to receive as the booster dose.

As mentioned in previous session, infection status of the vaccine recipients may also affect the immunogenicity of the vaccine. Our team have previously demonstrated that in COVID-19 recovered patients, a single dose of BNT162b2 induced a higher level of antibodies against SARS-CoV-2 variants than the inactivated virus vaccine [74]. Moreover, new generation vaccines that are based on the Omicron variant are now being developed and tested. One study using mice compares an Omicron-based mRNA booster against a wild-type-based-mRNA booster and shows that the strain-matched booster greatly enhanced neutralizing antibody titer against the Omicron variant, with comparable titers against the Delta variant [75]. The effect of an Omicron-based vaccine as a booster dose needs to be further studied. In conclusion, a wild-type-based mRNA vaccine can be considered for those who were previously vaccinated with non-mRNA vaccines and in those who were previously infected in view of enhanced protection against the Omicron variant. In addition, new generation Omicron-based vaccines may also be considered as booster dose vaccines.

## 5. When Should the Fourth Dose Booster Be Taken

The duration of vaccine protection after two doses of vaccine varies from 3 months to 6 months according to different studies, as summarized in Table 2 [74,76,77,78,79]. One of the reasons for this can be that these studies used different end points for measuring the effectiveness; some studies evaluated the level of serum-neutralizing antibodies after vaccination [77], some considered SARS-CoV-2 positivity rates in the community [76], and some studies investigated figures of hospitalization and death [78,79]. One drawback of these studies is that the majority of samples or analysis were carried out before the emergence of the Omicron variant. Since it was previously demonstrated that neutralizing activity against Omicron is non-comparable to other variants [8], the duration of protection after vaccination may be shorter than expected if we define protection using serum-neutralizing antibodies. On the other hand, the Omicron variant is associated with lower hospitalization rate and less severe disease, and so the duration of vaccine protection might be longer than expected if we use hospitalization rate and death as the definition of protection [80].

Recently, a study from Singapore demonstrated that after the third dose of mRNA vaccine, the serum level of antibodies against nucleocapsid protein and spike protein of the SARS-CoV-2 virus peaked at 30 days post-booster, persisted for at least 90 days, and was estimated to return to pre-booster levels after 8–11 months [81]. Studies from Israel evaluating the effectiveness of fourth-dose mRNA vaccine demonstrated that it had short-term benefits in reducing severe COVID-19, hospitalization and death in individuals who were aged 60 years or older, and had received the third dose for more than 4 months [82,83].

More studies are needed to define the duration of protection of a third booster dose, but in view of the results from two-dose vaccine studies and recent studies, a gap of 3 to 4 months after the third booster dose before administration of the fourth booster dose should be appropriate. The administration of the fourth dose shortly after the third dose may be unnecessary if the protective effect induced by the third dose has not waned off yet. Moreover, a previous study had shown that a delayed interval between the first two doses of the BNT162b2 mRNA vaccine, 8 to 16 weeks instead of the conventional 3 to 6 weeks, may induce better neutralizing against SARS-CoV-2 variants [84].

In addition, breakthrough infection in between vaccinations should also be considered, as these acute episodes can also augment the serum level of neutralizing antibodies. For instance, Omicron variant infection was demonstrated to enhance immunity against the Delta variant, but the Omicron virus can escape from a Delta variant infection-elicited immunity in both vaccinated and unvaccinated individuals [85]. To further complicate the scenario, the boosting effect of Omicron breakthrough infection is more obvious among vaccinated individuals with no previous infection. If the patient was previously infected, the boosting effect of Omicron is minimal [86]. In conclusion, a breakthrough infection may boost the serum-neutralizing antibodies which supports delaying the booster dose; however, virus strains of the infection may also be a contributing factor. More studies are required to delineate the duration of protection after a breakthrough infection to guide the interval between booster doses.

## 6. Discussion

The decision of whether a fourth dose should be mandatory is a balance between additional protection for people living in high-income countries and vaccine availability for the people living in developing areas. A fourth booster dose can transiently enhance immunity against virus variants of a particular population at the cost of delaying the immunization of those unvaccinated. Since the currently used vaccines were developed based on the wild-type virus and we are facing the emergence of SARS-CoV-2 variants, our review aims to raise the key questions (four Ws) on the protective benefits of a fourth booster dose, including the effect of the booster dose on virus variants (Why), susceptible groups of individuals who may benefit from additional booster doses (Who), the selection of vaccine platforms to better enhance immunity (What), and appropriate intervals between the third and the fourth doses (When). In general, a universal fourth booster dose may not be necessary; instead, after considering the four Ws, the decision of a fourth booster dose should be individualized. We propose that a fourth dose booster is appropriate after a few months for selected groups of individuals who have suboptimal protection against virus variants as a result of their health conditions or previous vaccination history. Moreover, bivalent vaccine boosters that target the Omicron variant are now authorized and currently undergoing investigation for their safety and immunogenicity; for immunocompetent individuals who plan to take the fourth-dose booster, it might be worthwhile to wait for the new bivalent vaccines, which will further increase the breadth of the variant’s coverage.

There are limitations with using in vitro testing of serum-neutralizing antibodies to guide vaccination policy. It was previously mentioned that the Omicron variant breakthrough infection boosted the level of serum antibodies in individuals who were not previously infected but not those who were previously infected with other strains [86]; however, in another study which compares individuals infected with BA.1 or BA (Omicron-primed) to those infected with pre-Omicron strains followed by BA.1 or BA.2 infection (double-primed), the double-primed cohort had a significantly lower incidence of reinfection during the period when BA.4 and BA.5 was predominant [87], with the concept of immune imprinting, exposure to different spike proteins during infection and vaccination can shape subsequent cross-protection against virus variants [88]. The protection of cell response elicited by Omicron infection may explain the discrepancy between in vitro antibody level and real-life incidence of reinfection. Besides, the cell-mediated immunity also plays a major role in the effectiveness of the COVID-19 vaccines. Overall, there is a lack of upper respiratory tract mucosal immunity stimulation from the current COVID-19 vaccines.

Aside from the protective effect of the additional booster, one should also consider the effect of leaving part of the world population unvaccinated. It was reported that while only 15% of the population in low-income countries had been vaccinated, wastage rates of COVID-19 vaccines was up to 30% [89]. We should never forget the history of effort to eradicate Polio; after the eradication of smallpox in 1979, the WHO had also established a global vaccination program in the hope of ending Polio. No new case of Polio was identified in the Western Pacific region and the European region by the end of the 1990s, and it was proposed that wild-type poliovirus would end by 2005 [90]; however, in 2016, there were still reports of cases of poliovirus type 1 in Afghanistan, Pakistan and Nigeria [91]. In recent years, there were reports of wild poliovirus importation to previously certified polio-free areas, leading to local outbreaks; for example, Syria, which has been polio-free since 1999, had a confirmed polio outbreak in 2014. The virus isolated was found to be the same strain as that which originated from Pakistan [92,93]. History has shown that if a virus is still active in part of the world, there is always a chance for it to re-emerge. Furthermore, our team had previously demonstrated that the SARS-CoV-2 virus mutates during the infection period [94], and unvaccinated hosts can potentially be a reservoir of new variant of concern. If this happens, the world will need to begin the global vaccination campaign again. Immunizing the neglected, unvaccinated population is vital in ending the COVID-19 pandemic. Proposed strategies to reduce wastage of vaccines includes measures to reduce overbooking vaccination appointments, reduce appointment-free vaccination, timely surplus donations and reallocations, as well as supply chain management in recipient countries [89]; in addition, regular stability testing of the current vaccines might be able to extend the shelf-life of these vaccines.

## 7. Conclusions

In conclusion, a fourth dose can temporarily boost the immune response against SARS-CoV-2 variants including Omicron; nevertheless, a universal fourth-dose strategy is not the solution to the current pandemic. A fourth dose can be considered for specific groups of individuals, such as the elderly, the immunocompromised, and previous vaccine platforms. The timing of the fourth dose should also be individualized and based on evidence from Israel studies that show a gap of 4 months after the third dose is appropriate. Most importantly, one should never overlook the important of vaccine equity as new variants will emerges if developing areas are neglected.

## Figures and Tables

**Figure 1 vaccines-10-01924-f001:**
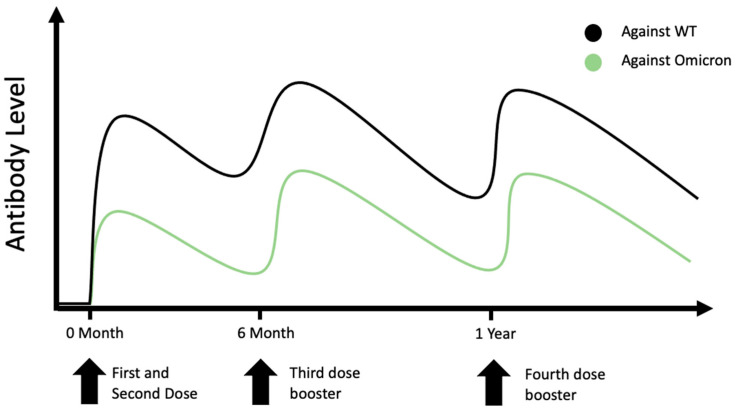
Changes in serum-neutralizing antibodies against SARS-CoV-2 Wild-Type virus and Omicron variant with time. Vaccine efficacy started to decline 2–4 months after the third-dose booster [13]. After third-dose booster, level of neutralizing antibodies against Omicron variant was lower than that against other strains such Wild-Type, Beta and Delta variants [8,15,16]. Furthermore, the response to the fourth-dose booster was comparable to the third-dose booster [22,23].

**Table 1 vaccines-10-01924-t001:** Immunogenicity of heterologous vaccine strategy against homologous vaccine strategy.

Study	Prime Dose	Booster Dose	Comparison	Strain Tested	Result
Clemens et al. [66]	CoronaVac	ChAdOx1 nCoV-19, BNT162b2, Ad26.COV2-s	CoronaVac	Delta, Omicron	Heterologous approach is superior
Stuart et al. [67]	ChAdOx1 nCoV-19 or BNT162b2 (one dose only)	mRNA1273, NVX-CoV2373 (one dose)	Same as Prime dose (one dose)	B.1.1.25	Heterologous second dosing with m1273 has transient systemic reactogenicity compared with homologous schedules
Khong et al. [8]	BNT162b2, CoronaVac	BNT162b2	CoronaVac	Wild-type, Beta variant, Delta variant, Omicron variant	BNT162b2 as booster in individuals primed with CoronaVac is comparable to three doses of BNT162b2 and superior to three doses of CoronaVac
Atmar et al. [68]	mRNA-1723, Ad26.COV2.S, BNT162b2	Vaccine of different platform	Vaccine of Same platform	D416G variant, Delta variant, Beta variant	Heterologous prime-boost strategies may extend the breadth and longevity of protection
Li et al. [69]	Coronavac	AD5-nCOV	CoronaVac	Wild-type, Delta variant	Heterologous boosting is more immunogenic

**Table 2 vaccines-10-01924-t002:** Duration of protection induced by two doses of COVID-19 vaccine.

Study	Vaccine Used	End-Point	Variant Tested	Duration of Protection
Erice et al. [77]	BNT162b2	Anti-N antibodies	N/A	3 months
Menni et al. [76]	ChAdOx1 nCov19, BNT162b2, mRNA1273	SARS-CoV-2 positivity rate	N/A	5 months
Andrews et al. [78]	ChAdOx1-S, BNT162b2	COVID-19-related hospitalization and death	B.1.1.7 (alpha), B.1.617.2 (delta)	20 weeks (4.5 months)
Nordström et al. [79]	ChAdOx1 nCoV-19, mRNA1273, BNT162b2	SARS-CoV2 infection of any severity, hospitalization and all-cause 30-day mortality after confirmed infection	N/A	4 months
Zhang el at. [74]	BNT162b2, Sinovac	Neutralizing antibodies	Wild-Type, Delta variant	6 months

## Data Availability

Not applicable.

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
