# Peer review of "The Four Ws of the Fourth Dose COVID-19 Vaccines: Why, Who, When and What"

_vaccines, 2022, doi:10.3390/vaccines10111924_

Round 1

Reviewer 1 Report

In chapter 5 (line 171-202), it would be very helpful if not only the time interval between the 3rd and 4th vaccination and vaccines, but also to an infection (delta-variant) or Omicron variant were discussed.

Author Response

In chapter 5 (line 171-202), it would be very helpful if not only the time interval between the 3rd and 4th vaccination and vaccines, but also to an infection (delta-variant) or Omicron variant were discussed.

Response: We thank Reviewer 1 for the comment, discussion on timing of booster dose and breakthrough infection was added.

Line 242-261.

“In addition, breakthrough infection in between vaccinations should also be con-sidered, as these acute episodes can also augment serum level of neutralizing antibodies. For instance, Omicron variant infection was demonstrated to enhance immunity against Delta variant but Omicron virus can escape from a Delta variant infection-elicited immunity in both vaccinated and unvaccinated individuals (85). To further complicate the scenario, the boosting effect of Omicron breakthrough infection is more obvious among vaccinated individuals with no previous infection. If the patient was previously infected, the boosting effect of Omicron is minimal (86). In conclusion, a breakthrough infection may boost the serum neutralizing antibody which supports delaying the booster dose, however, virus strains of the infection may also be a contributing factor. More study is required to de-lineate the duration of protection after a breakthrough infection to guide the interval between booster doses.”

Reviewer 2 Report

Article: The 4 Ws of the fourth dose COVID-19 vaccines: Why, Who, When and What

Journal: Vaccines

Type: Review

The review discussed four topics related to the fourth dose of COVID-19 vaccines: (Why) effectiveness of booster dose against virus variant, (Who) susceptible groups of individuals who may benefit from additional booster dose, (What) selection of vaccine platform to better enhance immunity and (When) appropriate interval between the third and fourth booster dose.

Introduction: The topic presented a brief description of the review. In my opinion, it is perfect to contextualize the reader about the importance of the study.

Why: The authors could include more information about the epidemiological aspects associated with a third dose (population data) because some countries included the booster dose in their vaccination plans. In addition, the authors could include a figure presenting the vaccine effects and improvement of the immunity response based on a fourth dose of the COVID-19 vaccines to improve the understanding of the manuscript.

Who: In the following sentence “3 unique groups of people were identified, including elderlies, immunocompromised patients, and individuals who were previously infected and vaccinated.” How the authors identified those groups. Also, I believe other groups deserve attention, for example, people with difficulties accessing health support services and Indigenous peoples. 

Also, in my opinion, it is important to include a table or a figure summarizing the information. In addition, the authors should include a statement about the particularities of each country and its population regarding the priority of vaccination against COVID-19.

What: No comments

When: No comments

Discussion: This topic should be better explored mainly about the importance of the fourth dose in a world scenario. In addition, it is important to discuss the strategies to optimize vaccination against SARS-CoV-2 to reduce the chance of an increase in the number of cases and death again. Also, the authors can discuss (briefly) the movement against vaccination that occurred in several countries and the political issues involved in the vaccination against the disease worldwide. Finally, if possible, the importance of the 4Ws should be discussed and contextualized in terms of applicability.

Conclusion: To reformulate based on the 4Ws. In this context, the authors can include four minor sentences finishing with the brilliant statement: “Most importantly, one should never overlook the importance of vaccine equity as new variants will emerge if the developing areas are neglected.”

Author Response

The review discussed four topics related to the fourth dose of COVID-19 vaccines: (Why) effectiveness of booster dose against virus variant, (Who) susceptible groups of individuals who may benefit from additional booster dose, (What) selection of vaccine platform to better enhance immunity and (When) appropriate interval between the third and fourth booster dose.

Introduction: The topic presented a brief description of the review. In my opinion, it is perfect to contextualize the reader about the importance of the study.

Why: The authors could include more information about the epidemiological aspects associated with a third dose (population data) because some countries included the booster dose in their vaccination plans. In addition, the authors could include a figure presenting the vaccine effects and improvement of the immunity response based on a fourth dose of the COVID-19 vaccines to improve the understanding of the manuscript.

Response: We thank Reviewer 2 for the comment, epidemiological aspect of booster dose was added to the manuscript.

Line 50-54

“Currently, a third dose booster is universal in some part of the world but not the others, up to 31st October, 2022, doses of COVID-19 vaccine boosters administered is highest in Chile, with 139.3 doses of booster dose was given per 100 individuals, the number is lower in India and Russia, with 15.5 doses and 13.2 doses of booster dose given per 100 individuals. Overall, the world average doses of booster per 100 people is 32.5 (11).”

Figure 1 is added to illustrate effect of booster doses (line 107-113) (Please see the attachment)

Who: In the following sentence “3 unique groups of people were identified, including elderlies, immunocompromised patients, and individuals who were previously infected and vaccinated.” How the authors identified those groups. Also, I believe other groups deserve attention, for example, people with difficulties accessing health support services and Indigenous peoples. 

Also, in my opinion, it is important to include a table or a figure summarizing the information. In addition, the authors should include a statement about the particularities of each country and its population regarding the priority of vaccination against COVID-19.

Response: We thank Reviewer 2 for the comment, the 3 groups of patients were selected because they may have different vaccine response to COVID-19 vaccine as compared with healthy individuals. We agree that people with difficulties accessing health support services needs more attention, nevertheless, we wish our review article can focuses more on whether a fourth dose booster should be considered as compared to current 3 doses regimen. Similarly, we believe discussion on particularities of each country and its population regarding the priority of vaccination against COVID-19 maybe too complicated to be included in this section.

The sentence was modified for clarification.

line 116-119.

3 unique groups of people are of interest, including elderlies, immunocompromised patients and individuals who were previously infected with SARS-CoV-2.

Figure 2 is included to summarize the information (line 165-173)(Please see the attachment)

Discussion: This topic should be better explored mainly about the importance of the fourth dose in a world scenario. In addition, it is important to discuss the strategies to optimize vaccination against SARS-CoV-2 to reduce the chance of an increase in the number of cases and death again. Also, the authors can discuss (briefly) the movement against vaccination that occurred in several countries and the political issues involved in the vaccination against the disease worldwide. Finally, if possible, the importance of the 4Ws should be discussed and contextualized in terms of applicability.

Response: We thank Reviewer 2 for the comment, we have modified the discussion accordingly.

Line 264 – 320.

“The decision of whether a fourth dose should be mandatory is a balance between additional protection to people living in high-income countries and vaccine availability to the people living in the developing areas. A fourth booster dose can transiently enhance immunity against virus variants of a particular population as the cost of delaying immunization of those unvaccinated. Since currently used vaccine were developed base on wild type virus and we are facing emergence of SARS-CoV-2 variants, our review aims to raise the key questions (4 Ws) on the protective benefits of a fourth booster dose, including effect of the booster dose on virus variant (Why), susceptible group of individuals who may benefits from additional booster dose (Who), selection of vaccine platform the better enhance immunity (What) and appropriate interval between the third and the fourth dose (When). In general, universal fourth booster dose may not be necessary, instead, after considering the 4Ws, the decision of a fourth booster dose should be individualized. We propose that a fourth dose booster is appropriate after a few months for selected group of individuals who have suboptimal protection against virus variants as result of their health condition or previous vaccination history. Moreover, bivalent vaccine boosters that target the Omicron variant are now authorized and currently undergoing investigation for their safety and immunogenicity, for immunocompetent individuals who plans to take the fourth dose booster, it might be worthwhile to wait for the new bivalent vaccines, which will further increase the breadth of the variant’s coverage.

There are limitations with using in vitro testing of serum neutralizing antibody to guide vaccination policy. It was previously mentioned that Omicron variant break-through infection boost level of serum antibody in individuals who were not previously infected but not those who were previously infected with other strains (86), however, in another study which compares individuals infected with BA.1 or BA.2 (omicron-primed) to those infected with pre-omicron strains followed by BA.1 or BA.2 infection (double-primed), the double-primed cohort has a significant lower incidence of reinfection during the period when BA.4 and BA.5 was predominant (87), with the concept of immune imprinting, exposure to different spikes proteins during infection and vaccination can shape subsequent cross-protection against virus variants (88). The protection of cell response elicited by Omicron infection may explain the discrepancy between in vitro antibody level and real-life incidence of reinfection. Besides, the cell mediated immunity also play a major role in the effectiveness of the COVID-19 vaccines. Overall, there is a lack of upper respiratory tract mucosal immunity stimulation from the current COVID-19 vaccines.

Aside from the protective effect of the additional booster, one should also consider the effect of leaving part of the world population unvaccinated, it was reported that while only 15% of the population in low-income countries had been vaccinated, wastage rates of COVID-19 vaccines was up to 30% (89). We should never forget the history of effort to eradicate Polio, after eradication of smallpox in 1979, WHO had also established global vaccination program in the hope of ending Polio. No new case of Polio was identified in Western Pacific region and European region by the end of 1990s and it was proposed that wild type poliovirus would end by 2005 (90), however, in 2016, there were still report of cases of poliovirus type 1 in Afghanistan, Pakistan and Nigeria (91). In recent years, there were report of wild poliovirus importation to previously certified polio-free areas, leading to local outbreaks, for example, Syria, which has been polio-free since 1999, had a con-firmed polio outbreak in 2014, the virus isolated was found to be the same strain as that originated from Pakistan (92, 93). History had shown that if a virus is still active in part of the world, there is always a chance for it to re-emerge. Furthermore, our team had previously demonstrated that SARS-CoV-2 virus mutates during the infection period (94), and unvaccinated hosts can potentially be reservoir of new variant of concern. If this happens, the world will need to start over the global vaccination campaign. Immunizing the neglected, unvaccinated population is vital in ending the COVID-19 pandemic. Proposed strategies to reduce wastage of vaccines includes measures to reduce over-booking vaccination appointments, reduce appointment-free vaccination, timely surplus donations and reallocations, as well as supply chain management in recipient countries (89), in addition, regular stability testing of the current vaccines might be able to extend the shelf-life of these vaccines.”

Response: However, we think discussion on anti-vaccination campaign and other political issue maybe out of the scope of our current review.

Conclusion: To reformulate based on the 4Ws. In this context, the authors can include four minor sentences finishing with the brilliant statement: “Most importantly, one should never overlook the importance of vaccine equity as new variants will emerge if the developing areas are neglected.”

Response: We thank Reviewer 2 for the comment.

Reviewer 3 Report

Authors are discussing a very timely topic of booster COVID-19 vaccines.  In general, authors lay out most of the main issues regarding booster vaccines quite well.  I have minor suggestions to the manuscript

I would suggest mentioning about immune imprinting as a reason for lower number of neutralizing antibodies. 

Unreported mild or asymptomatic COVID-19, which may hamper the final conclusions of many studies that are done to compare vaccine efficacy.  Due to home-testing and low reporting of COVID-19 infections it is difficult to know which ones of the vaccinees have also had COVID-19 infection.

Authors should include discussion of the new variants BQ1 and BB and the vaccine effectiveness.

Line 88: I would include a mention of the dose of the vaccine.  As an example Moderna vaccine dose is higher than Phizer mRNA dose.

Line 103: It is unclear to me why ‘individuals who were previously infected and vaccinated’ belong to a special group.  I would rather see a mention about the Long COVID individuals.  Currently we do not know who are at risk for Long COVID.

Line 198-201.  These two sentences are not adding to the review and are unclear.  I would suggest deleting them.

Line 204: Authors should point out the difficulty to turn a vaccine into vaccination in many countries in the world.  Many vaccines doses expire and need to be disposed.

Author Response

Authors are discussing a very timely topic of booster COVID-19 vaccines.  In general, authors lay out most of the main issues regarding booster vaccines quite well.  I have minor suggestions to the manuscript

I would suggest mentioning about immune imprinting as a reason for lower number of neutralizing antibodies. 

Unreported mild or asymptomatic COVID-19, which may hamper the final conclusions of many studies that are done to compare vaccine efficacy.  Due to home-testing and low reporting of COVID-19 infections it is difficult to know which ones of the vaccinees have also had COVID-19 infection.

Authors should include discussion of the new variants BQ1 and BB and the vaccine effectiveness.

Response: We thank Reviewer 3 for the comment, we have modified the discussion section accordingly.

Line 283-297.

“There are limitations with using in vitro testing of serum neutralizing antibody to guide vaccination policy. It was previously mentioned that Omicron variant break-through infection boost level of serum antibody in individuals who were not previously infected but not those who were previously infected with other strains (86), however, in another study which compares individuals infected with BA.1 or BA.2 (omicron-primed) to those infected with pre-omicron strains followed by BA.1 or BA.2 infection (dou-ble-primed), the double-primed cohort has a significant lower incidence of reinfection during the period when BA.4 and BA.5 was predominant (87), with the concept of immune imprinting, exposure to different spikes proteins during infection and vac-cination can shape subsequent cross-protection against virus variants (88). The protection of cell response elicited by Omicron infection may explain the discrepancy between in vitro antibody level and real-life incidence of reinfection. Besides, the cell mediated immunity also play a major role in the effectiveness of the COVID-19 vaccines. Overall, there is a lack of upper respiratory tract mucosal immunity stimulation from the current COVID-19 vaccines.”

Response: The new variants BQ1 and BB and vaccine effectiveness is not discussed because lack of published data.

Line 88: I would include a mention of the dose of the vaccine.  As an example Moderna vaccine dose is higher than Pfizer mRNA dose.

Response: We thank Reviewer 3 for the comment, and we have add a few sentence of discuss the effect of dosage of vaccines.

Line 190-198

“Furthermore, dosage of the booster dose may also affect the vaccine response, studies suggested that mRNA-1273 vaccine has a higher immunogenicity and effectiveness as compared to BNT162b2 vaccine (72, 73), using mRNA-1273 as the third dose booster was also demonstrated to be superior than BNT162b2 in individuals primed with BNT162b2 vaccine (67). One possible explanation could be a higher vaccine dose, with one dose of mRNA-1273 contains 100 μg of active substance while one dose of BNT162b2 vaccine contains 30 μg of active ingredient (61, 62). The dosage of the vaccine may be another consideration when considering what vaccine to receive as the booster dose.”

Line 103: It is unclear to me why ‘individuals who were previously infected and vaccinated’ belong to a special group.  I would rather see a mention about the Long COVID individuals.  Currently we do not know who are at risk for Long COVID.

Response: We thank Reviewer 3 for the comment. Similar to the concept of immune imprinting, we believe individuals who were infected, when compared to SARS-CoV-2 naïve individuals, may have differences in serum level and repertoire of antibodies against SARS-CoV-2. The wording may be misleading so we rephrased it.

Line116-118.

“3 unique groups of people are of interest, including elderlies, immunocompromised patients and individuals who were previously infected with SARS-CoV-2.”

We agree that Long COVID patients is an important group, and have suggested impact of vaccination on long covid patients.

Line157-163

“Interestingly, vaccination may also benefit patients with long COVID symptoms, as systemic reviews had suggested that vaccination before SARS-CoV-2 infection can reduce incidence of long COVID symptoms, while vaccination after an acute infection may also alleviate the long COVID symptoms (58, 59), however, these data were based on individuals who received two doses of vaccine. More data is needed to determine whether individuals who were infected may benefit from a fourth dose booster in terms of protection and long COVID symptoms.”

Line 198-201.  These two sentences are not adding to the review and are unclear.  I would suggest deleting them.

Response: We thank Reviewer 3 for the comment, the sentences have been removed.

Line 204: Authors should point out the difficulty to turn a vaccine into vaccination in many countries in the world.  Many vaccines doses expire and need to be disposed. (Discussion)

Response: We Thank Reviewer 3 for the comment, we have added a few sentence to discuss vaccine wastage and measures to reduce wastage of vaccines in discussion section.

Line 298-320

“Aside from the protective effect of the additional booster, one should also consider the effect of leaving part of the world population unvaccinated, it was reported that while only 15% of the population in low-income countries had been vaccinated, wastage rates of COVID-19 vaccines was up to 30% (89). We should never forget the history of effort to eradicate Polio, after eradication of smallpox in 1979, WHO had also established global vaccination program in the hope of ending Polio. No new case of Polio was identified in Western Pacific region and European region by the end of 1990s and it was proposed that wild type poliovirus would end by 2005 (90), however, in 2016, there were still report of cases of poliovirus type 1 in Afghanistan, Pakistan and Nigeria (91). In recent years, there were report of wild poliovirus importation to previously certified polio-free areas, leading to local outbreaks, for example, Syria, which has been polio-free since 1999, had a con-firmed polio outbreak in 2014, the virus isolated was found to be the same strain as that originated from Pakistan (92, 93). History had shown that if a virus is still active in part of the world, there is always a chance for it to re-emerge. Furthermore, our team had previously demonstrated that SARS-CoV-2 virus mutates during the infection period (94), and unvaccinated hosts can potentially be reservoir of new variant of concern. If this happens, the world will need to start over the global vaccination campaign. Immunizing the neglected, unvaccinated population is vital in ending the COVID-19 pandemic. Proposed strategies to reduce wastage of vaccines includes measures to reduce over-booking vaccination appointments, reduce appointment-free vaccination, timely surplus donations and reallocations, as well as supply chain management in recipient countries (89), in addition, regular stability testing of the current vaccines might be able to extend the shelf-life of these vaccines.”